# Client-care provider interaction during labour and birth as experienced by women: Respect, communication, confidentiality and autonomy

**Marit S. G. van der Pijl**[1]\*, **Marlies Kasperink**[1], **Martine H. Hollander**[2], **Corine Verhoeven**[1,3,4], **Elselijn Kingma**[5,6], **Ank de Jonge**[1]

**1** Amsterdam University Medical Centre (UMC), Vrije Universiteit Amsterdam, Department of Midwifery Science, AVAG/Amsterdam Public Health, Amsterdam, The Netherlands, **2** Amalia Children's Hospital, Department of Obstetrics, Radboud University Medical Centre, Nijmegen, The Netherlands, **3** Division of Midwifery, School of Health Sciences, University of Nottingham, Nottingham, United Kingdom, **4** Department of Obstetrics and Gynaecology, Maxima Medical Centre, Veldhoven, The Netherlands, **5** Department of Philosophy, University of Southampton, Southampton, United Kingdom, **6** Department of Industrial Engineering & Innovation Sciences, Philosophy & Ethics, Technical University Eindhoven, Eindhoven, The Netherlands

\* m.vanderpijl@amsterdamumc.nl

## Abstract

### Introduction

Respectful Maternity Care is important for achieving a positive labour and birth experience. Client-care provider interaction—specifically respect, communication, confidentiality and autonomy—is an important aspect of Respectful Maternity Care. The aim of this study was twofold: (1) to assess Dutch women's experience of respect, communication, confidentiality and autonomy during labour and birth and (2) to identify which client characteristics are associated with experiencing optimal respect, communication, confidentiality and autonomy.

### Methods

Pregnant women and women who recently gave birth in the Netherlands were recruited to fill out a validated web-based questionnaire (ReproQ). Mean scores per domain (scale 1–4) were calculated. Domains were dichotomised in non-optimal (score 1, 2,3) and optimal client-care provider interaction (score 4), and a multivariable logistic regression analysis was performed.

### Results

Of the 1367 recruited women, 804 respondents completed the questionnaire and 767 respondents completed enough questions to be included for analysis. Each domain had a mean score above 3.5. The domain confidentiality had the highest proportion of optimal scores (64.0%), followed by respect (53.3%), communication (45.1%) and autonomy (36.2%). In all four domains, women who gave birth at home with a community midwife had a higher proportion of optimal scores than women who gave birth in the hospital with a

**Funding:** The author(s) received no specific funding for this work.

**Competing interests:** The authors have declared that no competing interests exist.

(resident) obstetrician or hospital-based midwife. Lower education level, being multiparous and giving birth spontaneously were also significantly associated with a higher proportion of optimal scores in (one of) the domains.

## Discussion

This study shows that on average women scored high on experienced client-care provider interaction in the domains respect, communication, confidentiality and autonomy. At the same time, client-care provider interaction in the Netherlands still fell short of being optimal for a large number of women, in particular regarding women's autonomy. These results show there is still room for improvement in client-care provider interaction during labour and birth.

## Introduction

Respectful Maternity Care (RMC)—an approach to care that focuses on respecting the rights of women, newborns and their families to receive evidence based care while taking into account their personal needs and preferences—is a global priority, as also stated by the WHO in its latest recommendations for intrapartum care [1]. By expanding the focus of maternity care beyond the prevention of morbidity and mortality to encompass respect for women's autonomy, dignity, feelings and choices, optimal care can be provided and a positive labour and birth experience can be achieved [1, 2].

Women's negative experiences of labour and birth have been widely investigated in the literature [3–6]. Rijnders et al. (2008) showed that, after three years, 16% of women look back negatively on their birth in the Netherlands [7]. Soet, Brack and Dilorio (2003) found that negative interactions with health care providers are significant predictors of a negative perception of the birth experience [8]. Furthermore, Grekin and O'Hara (2014) revealed that hostile or disrespectful interactions can lead to experiencing birth as traumatic, and were even found to be associated with postpartum Post Traumatic Stress Disorder(PTSD) [9, 10]. This suggests that the way women are cared for plays an important role in women's labour and birth experiences [11–13].

Freedman and Kruk (2014) described disrespectful care as 'interactions or facility conditions that local consensus deem to be humiliating or undignified, and those interactions or conditions that are experienced as or intended to be humiliating or undignified' [14].

Previous studies have reported that different groups of women experience differences in care provisions. Vedam et al. (2019) found that one in six women experience mistreatment, with women of colour experiencing more disrespectful care by health care providers during labour and birth compared to white women [15]. A younger age at birth has also been found to be related to more experienced mistreatment of women during birth [16]. Women with a lower level of education experience less respectful care by their health care providers than women with a higher level of education, indicating a difference in experienced maternity care depending on socioeconomic status or level of knowledge [16, 17]. These signs of differences in experiences emphasise the need to address inequalities and to promote RMC for all women to achieve a positive labour and birth experience [16].

RMC emphasises underlying professional ethics, psycho-socio-cultural aspects of health care delivery and patient centredness as essential elements of labour and birth care [1, 2]. In 2011, Respectful Maternity Care rights were established which are based on international or

multinational human right instruments. The formulated rights serve as an important account-ability tool for recognising and protecting human rights of women during childbirth world-wide, as they are superior to cultural norms in society that enable disrespect and abuse to occur [1, 18, 19].

Good communication and a trusting and respectful relationship between women and their health care providers contribute to a positive labour and birth experience, and are even found to be more important factors than management of pain, pain relief and medical interventions [6, 20, 21]. These findings substantiate the importance of good client-care provider interaction during labour and birth, making it interesting to zoom in on this specific element of care [3, 21].

Although the literature does not provide one clear definition of interaction, it can be described as a process of cognition and action, in which the actions can be physical acts, acts of interplay or contact of verbal or nonverbal communication [22]. In terms of general health care, Donabedian (2003) described that client-care provider interaction has significant effects on patients perceptions of the quality of care and is therefore an important aspect of care to assess [23]. This is also emphasised by the WHO, who developed the responsiveness concept in 2000 to measure quality of care from a client's perspective, taking into account respect for human dignity and interpersonal aspects of the care process. The responsiveness concept covers eight domains, of which four focus on interactions with health care providers; dignity and respect, communication, confidentiality and autonomy [24]. Focusing on the aspect of interaction, rather than on childbirth experience as a whole, can provide more detailed insight into the experienced quality of client-care provider interaction, which can be helpful to inform and warrant implementation of RMC practices [25].

The four interaction domains of the responsiveness concept each cover different aspects of interaction. Dignity and respect cover receiving care in a respectful, caring, non-discrimina-tory setting. This includes aspects such as politeness, greeting and personal attention. Communication focuses on all types of contacts between the population and the health system, specifically care providers listening carefully to the concerns of the patient and explaining information with care. Other aspects of communication are the avoidance of non-technical language, frequencies of smiles, eye contact and voice quality. Confidentiality covers the privacy of the patient and the confidentiality of medical records and personal information. Autonomy can be divided into four themes: the need to provide information to individuals about their health status and risks; the need to involve individuals in the decision-making process to the extent that they wish; the need to obtain informed consent and the right of patients to refuse treatment [26].

The current study investigates the four domains of client-care provider interaction during labour and birth in the Netherlands from the client's point of view. The aim of this study was twofold: (1) to assess Dutch women's experience of dignity and respect, communication, confidentiality and autonomy during labour and birth and (2) to identify which characteristics of women and her pregnancy and birth are associated with experiencing optimal dignity and respect, communication, confidentiality and autonomy.

## Methods

### Study setting

The Dutch maternity care system is divided in primary midwife-led care and secondary obstetrician-led care. Women with a low risk pregnancy receive care from community midwives in a primary health care setting and give birth either at home, in a birth centre or in a hospital with their own midwife. Women with risk factors or complications receive secondary

obstetrician-led care in the hospital, where they are cared for by hospital-based midwives or (resident) obstetricians [27]. Most European countries report fewer than 1% homebirths and the majority of births take place in the hospital with secondary obstetrician-led care [28]. In the Netherlands, 50% of women start labour in primary care, 13% of all births take place at home assisted by a community midwife and 17% in a birth centre or the hospital assisted by a community midwife. 70% of the births take place in the hospital assisted by a hospital-based midwife or (resident) obstetrician [29].

To improve the quality of maternity care in the Netherlands, interprofessional and inter-organisational collaboration between primary midwife-led care and secondary obstetrician-led care has intensified in the last couple of years. This has led to the development of the "Standard for Integrated Birth Care" ("Zorgstandaard Integrale Geboortezorg"), developed by the National Health Care Institute in 2016 [30]. Integrated maternity care is defined as a closer collaboration between maternity care providers in order to put the women's needs and preferences at the centre of care.

## Study design

In the 'INtegrated CAre System'(INCAS) study, barriers and facilitators for integration of care during labour in the Netherlands were examined. A follow up study, the INtegrated CAre System study 2 (INCAS-2), aimed to evaluate the quality of integrated maternity care [31, 32]. As part of the quality of care assessment, women's experiences of labour and birth were investigated. The current study is a secondary analysis of the data collected in the INCAS-2 study. The data of interest consisted of a sample of Dutch-speaking women who gave birth between 2015–2018 and shared their experiences through a web-based validated questionnaire (ReproQ) [33].

## Ethical approval and informed consent

The medical ethics committee of Amsterdam UMC decided that the INCAS-2 study did not require ethics approval in the Netherlands (METC, VU University Medical Centre, no. 2014.160, 17th of April 2014). All respondents signed an informed consent form in which they gave consent to receive an invitation by email for an online questionnaire after they had given birth.

## Study population and sampling techniques

In four regions in the Netherlands, women were recruited between the first of January 2016 and the 31st of December 2017. Midwives from 30 midwifery practices handed out the informed consent form to all women in their practice between 38 weeks gestation and one week postpartum, asking women to participate in the study. All completed consent forms (including name, email address and signature) were sent to the researchers. The email addresses of women who gave consent were entered in a database. An email was sent to women two months after birth, in which they were requested to fill out the questionnaire. Respondents had a timeframe of four weeks to fill in the questionnaire. If necessary, two reminders were sent; the first one after eight days, the second one after 14 days. Respondents had to be at least 18 years old to be included in the study. As the questionnaire was only available in Dutch, women who could not read or write Dutch were excluded from the study.

## Measurement tools

Based on the WHO responsiveness concept [24], Scheerhagen et al. (2015) developed a questionnaire specifically for clients in Dutch maternity care, enabling direct measurement of

women's experiences of maternity care (ReproQ). The questionnaire was found suitable for assessing the quality of maternity care from the clients' perspective.

The ReproQ questionnaire consists of seven components: (1) the process of care, (2) the overall experience during birth and the postpartum period, (3) four domains on interactions with health care providers (dignity and respect, communication, confidentiality and autonomy), (4) four domains on experiences of the organisational setting (prompt attention, access to family and community support, quality of basic amenities, and choice and continuity of care), (5) the experienced health outcomes, (6) a ranking of the domains: respondents selected those two (out of eight) domains they felt were most important to them and (7) the respondents' socio-demographic characteristics.

The current study focuses on the four domains of the ReproQ that represent interactions with health care providers: dignity and respect (further referred to as 'respect'), communication, confidentiality and autonomy. The component on ranking the domains is included in the study to see how often the four domains on interaction were mentioned as important to the respondents. In the questionnaire, the respondents were asked to answer every question for two periods: 'during childbirth' and 'first week after childbirth'. Although optimal interaction with health care providers is important during any stage of pregnancy, labour, birth and the postpartum period, it is most relevant during labour and birth due to the intensity and vulnerability of this event [14]. Therefore, this study focused on the answers given for the option 'during childbirth'.

Table 1 provides an overview of the 17 questions in these four domains included in the analysis of this study. The response mode for the majority of the questions was a four point Likert scale, which included: 'never' (1), 'sometimes' (2), 'often' (3) and 'always' (4). Three questions had additional answer options, which are further described in the data analysis section.

## Data analysis

The completed questionnaires were imported into SPSS version 26 (IBM Corporation Inc. Armonk, NY, USA). Descriptive statistics were used to summarise the characteristics. In the

**Table 1. Questions of the domains that represent interactions with health care providers: Respect, autonomy, confidentiality and communication, for the setting 'during childbirth'.**

| ReproQuestionnaire domains | |
|---|---|
| Respect | 1. Did your healthcare providers take your privacy into account? <br> 2. Did your healthcare providers treat you with respect? <br> 3. Did you receive personal attention from your healthcare providers? <br> 4. Were your healthcare providers friendly? <br> 5. Did your healthcare providers take your wishes concerning your pregnancy and childbirth into account? <br> 6. Did you feel like you could tell everything to your health care providers? |
| Communication | 7. Did your healthcare providers answer your questions? <br> 8. Did your healthcare providers give the same advice? <br> 9. Did you understand the explanations the healthcare providers gave you? <br> 10. Did the healthcare providers tell you what was going to happen? |
| Confidentiality | 11. Did your health care providers discuss your medical situation with your family, only when you gave permission? <br> 12. Could you discuss important issues with your health care providers without others hearing it? <br> 13. Did your health care providers handle your medical files with care? |
| Autonomy | 14. Did you have input regarding your treatment? (this excludes emergency situations) <br> 15. Could you refuse a proposed treatment? (this excludes emergency situations) <br> 16. Did you have input regarding your pain treatment during childbirth? <br> 17. To what extent did you have influence on your birth plan? |

questionnaire, the respondents were asked about their personal characteristics: age (<25 (1),25-29(2), 30-34(3), 35-39(4), ≥40(5)), education level (low(1); primary school, first three years of secondary school or lower level of vocational training, medium(2); upper secondary school or higher vocational training, high(3): associate degree programs or doctoral degree programs [34]), self-identified ethnicity (Dutch(1), not Dutch(2)), and parity (primiparous(1), multiparous(2)). The following characteristics of the respondents' births were included: onset of labour (spontaneous(1), induction(2), caesarean section(3)), mode of birth (spontaneous vaginal birth(1), spontaneous vaginal birth with episiotomy(2), assisted vaginal delivery(3), planned caesarean section(4),unplanned caesarean section(5)) and place of birth (at home with community midwife(1), at the birth centre or hospital with community midwife(2), at the hospital with (resident) obstetrician or hospital-based midwife(3)). Characteristics were compared to the national perinatal data register of 2016 to determine whether the respondents were representative for the whole population of childbearing women in the Netherlands. The characteristic education level was compared to all women between 15–55 in 2016 registered by the Central Statistics Office (CBS) Netherlands (2016).

First, based on ranking the domains, percentages are given on how often respondents selected (one of) the presently included domains as most important to them, categorised as selected domain(1) and unselected domain(0).

Then, percentages and mean scores with standard deviations were calculated for all 17 questions from the four domains. Based on the mean scores of the individual questions per domain, a mean score for the overall domain was calculated. In case a respondent had 50% or more missing data in one of the four domains, she was excluded from the analysis for that particular domain.

After calculating mean scores, the domains were dichotomised in non-optimal interaction and optimal interaction. This dichotomisation leads to a stronger contrast between groups [35]. As the purpose of the current study was to determine the level of optimal client-care provider interaction, only 'always' was considered as optimal interaction, as this represents women who continuously felt they experienced optimal interaction during labour and birth. The answer options 'never' (1), 'sometimes' (2), and 'often' (3) were thus considered as non-optimal interaction and 'always' (4) was considered optimal interaction. If a respondent answered all questions within a domain with ´always´ (4), the complete domain was considered as optimal interaction. When one or more questions within a domain were answered with 'never' (1), 'sometimes' (2), or 'often' (3), the complete domain was considered as non-optimal interaction. When a domain included 'missings' <50%, the dichotomisation process was adjusted accordingly. For example, when a domain consisted of four questions, with one question classified as 'missing' and three as 'always', the domain was considered optimal.

Question 11 '*Did your health care providers discuss your medical situation with your family, only when you gave permission*?' had two additional answer options: (5) 'I don't know' and (6) 'not applicable'. Option (5) was considered missing, as the answer option does not provide an indication about the optimality of the interaction. Option (6) was considered optimal as this option states the situation had not taken place. Question 16 and 17 had different answer options than the above mentioned scale. Question 16, '*Did you have input regarding your pain treatment during childbirth*?', had the following answer options: (1) No, but I did not want to participate in the decision making process; (2) No, but I wanted to participate in the decision making process; (3) Yes, I partly decided; (4) Yes, I decided completely by myself; (5) Not applicable, for example because of a planned caesarean section; (6) Not applicable, the pain treatment was not discussed before giving birth. We considered options (1), (4) and (5) as optimal interaction as the respondent either did not wish to have input in their pain treatment, experienced full input in their pain treatment, or there was no possibility to choose a certain

pain treatment due to their situation). Options (2), (3) and (6) were considered non-optimal interaction as the respondent either felt not involved, partly involved or the pain treatment was not discussed prior to giving birth. The latter was categorised as non-optimal interaction, as the Dutch guideline for midwifery care states that care providers should discuss pain treatment with their clients during their pregnancy [30]. Question 17, *'To what extent did you have influence on your birth plan?'* had the following answer options: (1) I had no influence, without a medical reason; (2) I had little influence; (3) I had a lot of influence; (4) I decided completely by myself; (5) I had no influence, because of a medical reason; (6) It was not discussed. Options (3), (4), (5) were considered optimal, as the respondent either had a lot of influence, decided completely by themselves, or it was not applicable due to their situation. Options (1), (2) and (6) were considered non-optimal interaction either when the respondent had little to no influence on their birth plan or when the birth plan was not discussed. The Dutch guideline for midwifery care states that care providers should discuss the birth plan with their clients prior to birth [30]. Therefore, not discussing the birth plan was considered non-optimal interaction.

Percentages of the frequency of optimal score per domain were calculated. A univariable analysis was performed per domain and crude odds ratios (OR) with 95% confidence intervals (CI) were calculated. Possible determinants were age, educational level, self-identified ethnicity, parity, onset of labour, mode of birth and place of birth. A multivariable logistic regression analysis was performed with these variables being considered as independent variables and the four domains as the dependent variables. Adjusted odds ratios (AOR) with 95% CI per domain were calculated. Odds ratios above one indicated higher odds for optimal scores for the particular group compared to the reference group. An odds ratio of less than one indicated higher odds for non-optimal scores in a domain.

## Results

In total, 1367 women were recruited, of whom 804 (59%) women completed the questionnaire. Thirty-seven respondents were excluded because of 50% or more missing data in all included domains, leaving 767 respondents for analysis (Fig 1).

Table 2 shows the characteristics of the respondents compared to the Dutch national data of 2016. The largest group of respondents were between 30 and 35 years old at the time of giving birth (44.8%). The majority identified as of Dutch origin (87.7%). Over two thirds of the respondents had a high education level (68%). Most respondents had given birth for the first time (60.7%). Regarding the characteristics of the respondents' births, 674 had had a vaginal birth (88%) and 92 respondents had given birth by caesarean section (12%). Of all births, 171 took place at home with a community midwife (22.4%), 135 at a birth centre or similar place with a community midwife (17.6%) and 459 in the hospital with a (resident) obstetrician or hospital-based midwife (60%). Statistically significant differences were found for all of the characteristics of the respondents compared to all Dutch women who gave birth (p<0.001).

When asked to choose two of all eight domains as most important, 76.5% of the respondents selected at least one domain on interactions with health care providers. Comparing the eight individual domains, the domain respect was most often chosen as important by 53.2% of the respondents, followed by the domain autonomy, chosen by 28.8% of the respondents.

Table 3 presents the answers given to each question of the four domains on interactions with health care providers. The question 'Did your health care providers treat you with respect?' scored highest within the domain respect (mean 3.86 out of 4 [SD 0.41]), with 88.2% of respondents answering 'always', also being the highest scored question over all domains. Within the domain communication, the question 'Did your health care providers answer your questions?' scored highest (mean 3.79 [SD 0.49]), with 82.5% of the respondents answering

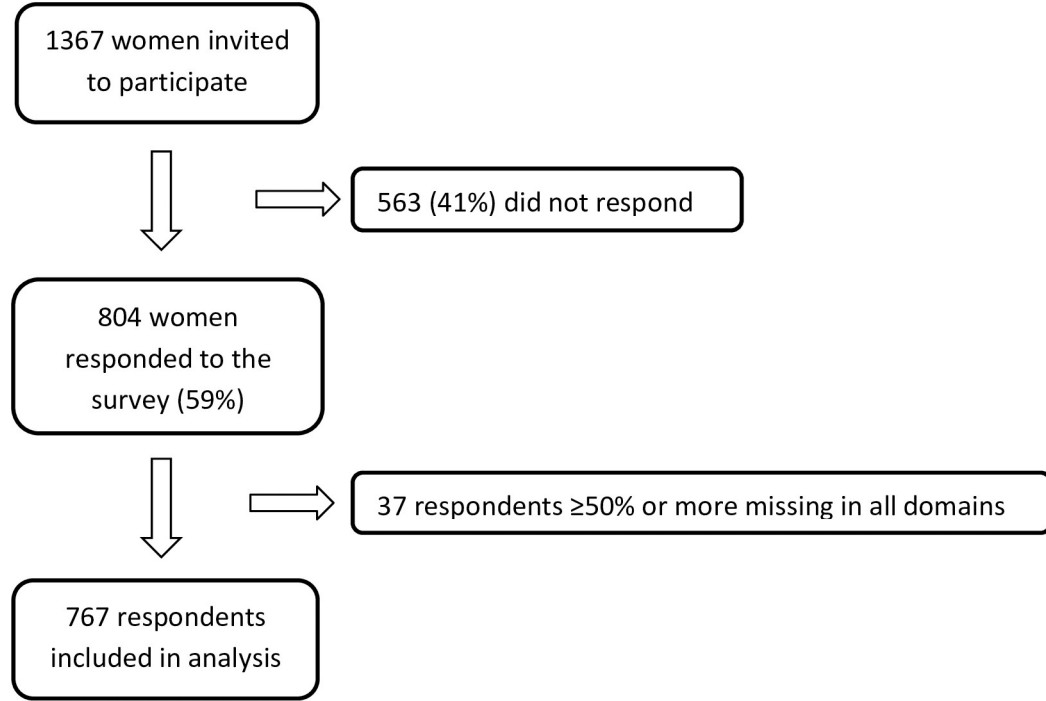

**Fig 1. Flowchart of women responding to the questionnaire (n = 767).**

'always'. 'Did your health care providers handle your medical files with care?' scored highest within the domain confidentiality, with 85.5% of the respondents answering 'always' (mean 3.82 [SD 0.50]). Within the domain autonomy, the question 'Did you have input regarding your treatment?' scored highest, with 71.5% of the respondents answering 'always' (3.63 [0.66]).

Table 3 also shows the overall means of the domains. The domain respect had the highest mean (3.77 [SD 0.39]), followed by the domain confidentiality (3.71 [SD 0.55]). When dividing the domains between optimal and non-optimal interaction, the domain confidentiality had the highest prevalence of optimal interaction (n = 491, 64%), followed by the domain respect (n = 409, 53.3%), the domain communication (n = 346, 45.1%) and the domain autonomy (n = 278, 36.2%).

Table 4 presents the association between respondents' characteristics and the prevalence of optimal interaction. These results will be discussed per domain.

## Domain respect

The univariable analysis revealed that age, education level, parity, onset of labour, mode of birth and place of birth were statistically significantly associated with experiencing respect (S1 Table). After adjusting for all variables, only the association with education level and place of birth remained statistically significant (Table 4). Compared to women with a low education level, women with a high education level had lower odds of experiencing respect (AOR 0.35 [CI 0.15–0.79]). Compared to respondents giving birth at home, the odds of experiencing respect were lower for respondents giving birth in a birth centre with the community midwife (AOR 0.53 [CI 0.31–0.90]) or at the hospital with a (resident) obstetrician or hospital-based midwife (AOR 0.31 [CI 0.19–0.51]).

**Table 2. Characteristics of the respondents (n = 767) compared to national data.**

| Characteristics | | n (%) or mean [SD] | | Chi-Square |
|---|---|---|---|---|
| | | Respondents | National perinatal registry 2016* | P-value*** |
| Age | Average | *31.7 [4.2]* | | p < 0.001 |
| | <25 | 30 (4) | 15881 (9.5) | |
| | 25–29 | 202 (26.4) | 51 348 (30.8) | |
| | 30–34 | 342 (44.8) | 63 772 (38.3) | |
| | 35–39 | 166 (21.7) | 30 128 (18.1) | |
| | ≥40 | 24 (3.1) | 5536 (3.3) | |
| | *Missing* | 3 | | |
| Self-identified ethnicity | Dutch | 619 (87.7) | 123 838 (74.3) | p < 0.001 |
| | Non-Dutch | 87 (12.3) | 42 856 (25.7) | |
| | *Antillean/Aruban* | *3 (0.4)* | | |
| | *Turkish/Kurdish* | *3 (0.4)* | | |
| | *Surinamese* | *17 (2.4)* | | |
| | *Moroccan* | *6 (0.8)* | | |
| | *Asian* | *20 (2.8)* | | |
| | *Eastern European* | *9 (1.3)* | | |
| | *Other* | *29 (4.1)* | | |
| | *Missing* | 61 | | |
| Education level | Low | 33 (4.7) | 1 073 000 (27.2)** | p < 0.001 |
| | Middle | 190 (27.3) | 1 748 000 (44.2)** | |
| | High | 473 (68.0) | 1 131 000 (28.6)** | |
| | *Missing* | 71 | | |
| Parity | Primiparous | 432 (60.7) | 73130 (43.9) | p < 0.001 |
| | Multiparous | 280 (39.3) | 93564 (56.1) | |
| | *Missing* | 55 | | |
| Onset of labour | Spontaneous | 599 (78.1) | 115755 (69.8) | p < 0.001 |
| | Induction | 139 (18.1) | 37644 (22.6) | |
| | Caesarean section | 29 (3.8) | 13448 (8.1) | |
| | *Unplanned* | *4 (0.5)* | | |
| Mode of birth | Spontaneous vaginal birth | 604 (78.9) | 126 613 (76.0) | p < 0.001 |
| | *with episiotomy* | *88 (11.5)* | | |
| | Assisted vaginal birth | 70 (9.1) | 13 417 (8.0) | |
| | Planned caesarean section | 24 (3.1) | 13 448 (8.1) | |
| | Unplanned caesarean section | 68 (8.9) | 13 216 (7.9) | |
| | *Missing* | 1 | | |
| Place of birth | At home with community midwife | 171 (22.4) | 21 434 (12.7) | p < 0.001 |
| | At a birth centre or hospital with community midwife | 135 (17.6) | 29 138 (17.3) | |
| | At the hospital with (resident) obstetrician or hospital-based midwife | 459 (60) | 117 868 (70.0) | |
| | *Missing* | 2 | | |

*Perined (2018). Perinatale zorg in Nederland 2016.

**based on all women between 15–55 in 2016 registered by CBS Statistics Netherlands (2016)

***significantly different from national data when p-value < 0.001

## Domain communication

After univariable analysis, age, education level, parity, onset of labour, mode of birth and place of birth were statistically significantly associated with experiencing optimal communication

**Table 3. Overview of questions of the domains respect, confidentiality, autonomy and communication with n (%) per answer given, n (%) of optimal care and mean [SD] per question and domain.**

| Domains and their questions | | n (%) | n (%) of optimal care | Mean [SD] |
|---|---|---|---|---|
| **Respect** | | | **409 (53.3)** | **3.77 [0.39]** |
| Did your health care providers take your privacy into account? | Never | 3 (0.4) | 601 (78.5) | 3.75 [0.52] |
| | Sometimes | 23 (3.0) | | |
| | Most of the times | 139 (18.1) | | |
| | **Always*** | **601 (78.5)** | | |
| | *Missing* | 1 | | |
| Did your health care providers treat you with respect? | Never | 1 (0.3) | 674 (88.2) | 3.86 [0.41] |
| | Sometimes | 12 (1.6) | | |
| | Most of the times | 76 (9.9) | | |
| | **Always** | **674 (88.2)** | | |
| | *Missing* | 3 | | |
| Did you receive personal attention from your health care providers? | Never | 3 (0.4) | 607 (79.6) | 3.76 [0.52] |
| | Sometimes | 24 (3.1) | | |
| | Most of the times | 129 (16.9) | | |
| | **Always** | **607 (79.6)** | | |
| | *Missing* | 4 | | |
| Were your health care providers friendly? | Never | 1 (0.1) | 655 (85.7) | 3.84 [0.43] |
| | Sometimes | 14 (1.8) | | |
| | Most of the times | 94 (12.3) | | |
| | **Always** | **655 (85.7)** | | |
| | *Missing* | 3 | | |
| Did your healthcare providers take your wishes concerning your pregnancy and childbirth into account? | Never | 8 (1.0) | 569 (74.4) | 3.67 [0.62] |
| | Sometimes | 39 (5.1) | | |
| | Most of the times | 149 (19.5) | | |
| | **Always** | **569 (74.4)** | | |
| | *Missing* | 2 | | |
| Did you feel like you could tell everything to your health care providers? | Never | 6 (0.8) | 595 (77.9) | 3.72 [0.58] |
| | Sometimes | 33 (4.3) | | |
| | Most of the times | 130 (17) | | |
| | **Always** | **595 (77.9)** | | |
| | *Missing* | 3 | | |
| **Communication** | | | **346 (45.1)** | **3.68 [0.43]** |
| Did your health care providers answer your questions? | Never | 3 (0.4) | 613 (82.5) | 3.79 [0.49] |
| | Sometimes | 18 (2.4) | | |
| | Most of the times | 109 (14.7) | | |
| | **Always** | **613 (82.5)** | | |
| | *Missing* | 24 | | |

*(Continued)*

**Table 3.** (Continued)

| Domains and their questions | | n (%) | n (%) of optimal care | Mean [SD] |
|---|---|---|---|---|
| Did your health care providers give the same advice? | Never | 11 (1.5) | 439 (59.3) | 3.5 [0.69] |
| | Sometimes | 48 (6.5) | | |
| | Most of the times | 242 (32.7) | | |
| | **Always** | **439 (59.3)** | | |
| | *Missing* | 27 | | |
| Did you understand the explanations the health care providers gave you? | Never | 2 (0.3) | 573 (77.0) | 3.75 [0.49] |
| | Sometimes | 13 (1.7) | | |
| | Most of the times | 156 (21) | | |
| | **Always** | **573 (77)** | | |
| | *Missing* | 23 | | |
| Did the health care providers tell you what was going to happen? | Never | 3 (0.4) | 550 (74.3) | 3.69 [0.57] |
| | Sometimes | 33 (4.5) | | |
| | Most of the times | 154 (20.8) | | |
| | **Always** | **550 (74.3)** | | |
| | *Missing* | 27 | | |
| **Confidentiality** | | | **491 (64.0)** | **3.71 [0.55]**\*\* |
| Did your health care providers discuss your medical situation with your family, only when you gave permission? \*\*\* | Never | 21 (3.1) | 365 (53.7) | |
| | Sometimes | 9 (1.3) | | |
| | Most of the times | 46 (6.7) | | |
| | **Always** | **365 (53.7)** | | |
| | **Not applicable** | **245 (35.7)** | | |
| | *Missing* | 81 (10.6) | | |
| Could you discuss important issues with your health care providers without others hearing it? | Never | 32 (4.3) | 549 (73.7) | 3.61 [0.76] |
| | Sometimes | 29 (3.9) | | |
| | Most of the times | 135 (18.1) | | |
| | **Always** | **549 (73.7)** | | |
| | *Missing* | 22 (2.9) | | |
| Did your health care providers handle your medical files with care? | Never | 8 (1.1) | 633 (85.5) | 3.82 [0.50] |
| | Sometimes | 12 (1.6) | | |
| | Most of the times | 87 (11.8) | | |
| | **Always** | **633 (85.5)** | | |
| | *Missing* | 27 | | |
| **Autonomy** | | | **278 (36.2)** | **3.57 [0.65]**\*\* |

(*Continued*)

**Table 3.** (Continued)

| Domains and their questions | | n (%) | n (%) of optimal care | Mean [SD] |
|---|---|---|---|---|
| Did you have input regarding your treatment? (this excludes emergency situations) | Never | 10 (1.3) | 541 (71.5) | 3.63 [0.66] |
| | Sometimes | 46 (6.1) | | |
| | Most of the times | 160 (21.1) | | |
| | **Always** | **541 (71.5)** | | |
| | *Missing* | 10 | | |
| Could you refuse a proposed treatment? (this excludes emergency situations) | Never | 31 (4.2) | 488 (65.9) | 3.51 [0.80] |
| | Sometimes | 51 (6.9) | | |
| | Most of the times | 170 (23) | | |
| | **Always** | **488 (65.9)** | | |
| | *Missing* | 27 | | |
| To what extent did you have influence on your birth plan?*** | I had no influence, without a medical reason | 6 (0.8) | 584 (76.1) | |
| | I had little influence | 28 (3.7) | | |
| | **I had a lot of influence** | **172 (22.7)** | | |
| | **I decided completely by myself** | **339 (44.7)** | | |
| | **I had no influence, because of a medical reason** | **73 (9.6)** | | |
| | It was not discussed | 141 (18.6) | | |
| | *Missing* | 8 | | |
| Did you have input regarding your pain treatment during childbirth?*** | **No, but I did not want to participate in the decision making process** | **6 (0.8)** | 518 (67.5) | |
| | No, but I wanted to participate in the decision making process | 17 (2.2) | | |
| | Yes, I partly decided | 123 (16.2) | | |
| | **Yes, I decided completely by myself** | **469 (61.8)** | | |
| | **Not applicable, for example because of a caesarean section** | **43 (5.7)** | | |
| | Not applicable, the pain treatment is not discussed before giving birth | 101 (13.3) | | |
| | *Missing* | 8 | | |

*answer options categorised as optimal care are shown in bold

**mean calculated based on questions in domain with four answer options

***no mean calculated due to divergent answer options

(S1 Table). After multivariable analysis, significant associations remained for three factors: respondents who were multiparous had higher odds of experiencing optimal communication compared to primiparous women (AOR 1.49 [CI 1.20–2.17,]). Compared to respondents who had a spontaneous birth, women who had a vaginal birth with episiotomy (AOR 0.56 [CI 0.33–0.95]) or an unplanned caesarean section (AOR 0.50 [CI 0.26–0.96]) had lower odds of experiencing optimal communication. Respondents who gave birth in the hospital with a (resi-dent) obstetrician or hospital-based midwife had lower odds of experiencing optimal

**Table 4. The association between characteristics and experiencing optimal respect, communication, confidentiality and autonomy: Outcomes of the multivariable logistic regression.**

| Characteristics | | Optimal interaction n (%) | Adjusted Odds Ratio [95% CI]* |
|---|---|---|---|
| **Respect** | | | |
| Education level | Low | 24 (72.7) | *ref*** |
| | Middle | 118 (62.1) | 0.50 [0.21–1.20] |
| | High | 235 (49.7) | **0.34 [0.15–0.79]** |
| Place of birth | At home with community midwife | 129 (75.4) | *ref* |
| | At the birth centre or hospital with community midwife | 79 (58.5) | **0.53 [0.31–0.90]** |
| | At the hospital with (resident) obstetrician or hospital-based midwife | 200 (43.6) | **0.31 [0.19–0.51]** |
| **Communication** | | | |
| Parity | Primiparous | 175 (40.5) | *ref* |
| | Multiparous | 159 (56.8) | **1.49 [1.20–2.17]** |
| Mode of birth | Spontaneous vaginal birth | 261 (50.6) | *ref* |
| | Vaginal birth with episiotomy | 30 (34.1) | **0.56 [0.33–0.95]** |
| | Assisted vaginal birth | 27 (38.6) | 0.99 [0.55–1.78] |
| | Planned caesarean section | 10 (41.7) | 0.82 [0.70–10.29] |
| | Unplanned caesarean section | 18 (26.5) | **0.50 [0.26–0.96]** |
| Place of birth | At home with community midwife | 105 (61.4) | *ref* |
| | At the birth centre or hospital with community midwife | 69 (51.1) | 0.73 [0.44–1.21] |
| | At the hospital with (resident) obstetrician or hospital-based midwife | 171 (37.3) | **0.52 [0.32–0.83]** |
| **Confidentiality** | | | |
| Place of birth | At home with community midwife | 136 (79.5) | *ref* |
| | At the birth centre or hospital with community midwife | 85 (63) | **0.47 [0.27–0.82]** |
| | At the hospital with (resident) obstetrician or hospital-based midwife | 269 (58.6) | **0.41 [0.24–0.70]** |
| **Autonomy** | | | |
| Mode of birth | Spontaneous vaginal birth | 212 (41.1) | *ref* |
| | Vaginal birth with episiotomy | 22 (25) | **0.56 [0.33–0.99]** |
| | Assisted vaginal birth | 22 (31.4) | 0.73 [0.40–1.35] |
| | Planned caesarean section | 6 (25) | 0.60 [0.05–7.59] |
| | Unplanned caesarean section | 16 (23.5) | 0.54 [0.28–1.04] |
| Place of birth | At home with community midwife | 80 (46.8) | *Ref* |
| | At the birth centre or hospital with community midwife | 57 (42.2) | 0.81 [0.49–1.33] |
| | At the hospital with (resident) obstetrician or hospital-based midwife | 141 (30.7) | **0.61 [0.38–0.98]** |

*Multivariable analysis adjusted for age, self-identified ethnicity, education level, parity, onset of labour, mode of birth and place of birth. Table only shows variables with at least one statistically significant result. Significant odds ratios are shown in bold.

**Reference category

communication compared to respondents who gave birth at home (AOR 0.52 [CI 0.32–0.83]) (Table 4).

## Domain confidentiality

Parity, mode of birth and place of birth were all found to be statistically significantly associated with optimal scores within the domain confidentiality (S2 Table). The multivariable analysis revealed only one statistically significant association: compared to respondents giving birth at home with a community midwife, the odds of optimal scores within the domain confidentiality was lower for respondents giving birth in a birth centre with the community midwife (0.47 [0.27–0.82]) or at the hospital with a (resident) obstetrician or hospital-based midwife (AOR 0.41 [0.24–0.70]) (Table 4).

### Domain autonomy

For the domain autonomy, parity, onset of labour, mode of birth and place of birth were all found to be statistically significantly associated with experiencing optimal autonomy after uni-variable analysis (S2 Table). After multivariable analysis, two variables remained significant. Compared to women with a spontaneous vaginal birth, women who gave birth with an episiotomy had lower odds of experiencing optimal autonomy (0.56 [0.33–0.99]). Place of birth also showed significant results on multivariable level: women who gave birth in the hospital with a (resident) obstetrician or hospital-based midwife had lower odds of experiencing optimal autonomy compared to women who gave birth at home with a community midwife (0.61 [0.38–0.98]) (Table 4).

## Discussion

In this study we investigated women's experiences of client-care provider interaction during labour and birth, measured through four domains: dignity and respect, communication, confidentiality and autonomy.

On a scale of 1–4, each domain had a mean score above 3.5. Dichotomisation in optimal and non-optimal interaction showed that the domain confidentiality had the highest rate of optimal scores (64.0%), followed by the domains respect (53.3%), communication (45.1%) and the domain autonomy (36.2%).

In all four domains, respondents who gave birth at home with a community midwife experienced a higher proportion optimal client-care provider interaction than respondents who gave birth in the hospital with a (resident) obstetrician or hospital-based midwife. Respondents with a low education level more frequently experienced optimal respect than respondents with a high education level. Respondents who were multiparous and respondents who gave birth spontaneously experienced a higher proportion of optimal communication than primiparous respondents and respondents who had an episiotomy or an unplanned caesarean section. Finally, respondents who gave birth spontaneously experienced optimal autonomy more frequently if they did not have an episiotomy compared to respondents who did.

Almost 90% of the respondents stated they were always treated with respect during labour and birth by their care providers, suggesting a high quality of experienced respectful care during labour and birth in the Netherlands. Baas et al. (2017) previously found that women in the Netherlands scored a mean of 3.9 for experienced patient centeredness of care providers during birth (scale 1–4) and van Stenus et al. (2018) showed that Dutch women rated their experience of perinatal health care with a mean score of 3.78 (scale 1–4) [36, 37]. The high mean scores on client-care provider interaction found in the current study are in line with above mentioned findings, indicating that the majority of labouring women experienced positive interactions with care providers.

Nevertheless, dichotomisation in optimal and non-optimal care showed a considerable number of women who did not experience optimal interaction during labour and birth. Recent studies from Italy, Australia and the US also reported the occurrence of non-optimal interaction and even mistreatment during labour and birth [11, 15, 38]. Vedam et al. (2019) conducted a study among postpartum women in the US and found that one in six women reported they experienced at least one type of mistreatment during labour and birth, which is considered an unacceptably high number, especially for a high resource setting [15]. Results of the current and previous studies underline that preventing mistreatment and achieving optimal client-care provider interaction during labour and birth is highly relevant, and attention on respectful care provision is needed to ensure a positive experience for all women. It is important to note that respectful care provision does not only entail the prevention of

mistreatment during care. Receiving Respectful Maternity Care is a universal human right that should be supported and promoted on several levels; individual, health facility, health system and in society [1, 39]. It is argued that in society, it is the standard to follow the rules of modern medicine in order to give birth according to protocol, whereby health care providers often have control over labour and birth [40]. This can lead to situations in which actions and interactions perceived by women as harmful are normalised and standardised. This not only jeopardises women's own labouring process, but also their role as a woman [41]. Therefore, achieving Respectful Maternity Care should include cultural changes in society, which in turn will affect the norms in which maternity care provision is carried out [18].

Women who gave birth vaginally with an episiotomy experienced optimal communication and autonomy less often than women who gave birth spontaneously without episiotomy. Research shows that selective episiotomy policies positively contribute to clinical outcomes [42, 43]. Although in the Netherlands there is a policy of selective use of episiotomy, compared to other countries such as Sweden and Finland, the percentage of episiotomies is relatively high [44]. Therefore, this finding is relevant for maternity care providers in the Netherlands and elsewhere where percentages are relatively high.

Although all domains left room for improvement, the domain autonomy had the lowest percentage of optimal interaction. Feijen et al. (2019) found a similar discrepancy between the level of experienced respect and autonomy in maternity care among pregnant women: 83% of women experienced a high quality of respect against 62% experiencing a high quality of autonomy [45]. This is reason for concern, because women's autonomy is important in maternity care; women who had a negative or traumatic birth experience often described not feeling seen or heard by care providers [13, 46, 47]. Beck (2004) described that women with a birth trauma often perceived their births being viewed as routine by care providers [48]. This may undermine a woman's role in the process, jeopardising her feeling of autonomy and control [11, 48]. Even when care providers actively safeguard clients' autonomous choices, they may face challenges such as time pressure, strict medical protocols and a work environment with a dominant biomedical framework [49]. These circumstances can create birth settings in which the woman is not the centre of care, which can explain the relatively low score of autonomy in the current study. Education and training for care professionals should emphasise the importance of women's autonomy and encourage debate about this subject within a professional setting, so that facilitators and barriers in respecting women's autonomy can be identified.

In all four domains women who gave birth at home with a community midwife experienced optimal interaction more frequently than women who gave birth in the hospital with a (resident) obstetrician or hospital-based midwife. It is important to note that the majority of the women who give birth assisted by a community midwife, compared to women who give birth assisted by a hospital based midwife or (resident) obstetrician), have an uncomplicated labour and birth. Complications or the need for medical interventions could require more extensive and difficult interaction between women and care providers, increasing the possibility for less optimal client-care provider interaction to occur during hospital-based births assisted by a hospital based midwife or (resident) obstetrician. However, the current study showed that non-optimal client-care provider interaction also occurred among women with a straightforward birth. This suggests there is room for improvement throughout the system; optimal client-care provider interaction should be perceived as an overarching element of care that should be guaranteed for all women during labour and birth, regardless of circumstances.

If we compare home births with births at a birth centre and hospital, it was found that women who gave birth at home experienced more respect and confidentiality compared to women giving birth in a birth centre or hospital. Studies from the US and the Netherlands show that women experience fewer unnecessary interventions and/or interruptions to the

birth process at home [50, 51]. Furthermore, women experience more freedom and comfort at home, are more active and feel more in control of their own birth [50]. Mondy et al. (2016) described that the increased feeling of control can be explained by women effortlessly taking ownership of the environment in their own home, compared to having to adapt to a new, unfamiliar setting. New settings can cause women to act and interact more passively with the environment, leading to lower quality of experienced control [52]. It is recommended that care providers inform women about this, so they can take this into account in their decision making regarding the preferred place of birth. In birth centres and hospitals, there should be a focus on ways to give women more ownership over the birthing environment in order to make them feel empowered and more in control when such a setting is wanted or needed.

Women who had an unplanned caesarean section were found to have experienced less optimal communication compared to women giving birth spontaneously. Previous literature showed that unplanned interventions during labour and birth, compared to planned interventions, were found to be more often associated with a negative childbirth experience [53]. Also, women undergoing unplanned caesarean section often have more complications compared to women giving birth by a planned caesarean section or vaginally [54, 55]. As complications require more extensive interaction, it is likely this influences women's experiences.

Jenkins et al. (2014) indicated that communication is one of the most important aspects of maternity care, especially for primiparous women [56]. In the current study, primiparous women experienced less optimal communication compared to multiparous women, which is also in line with previous studies [12, 57]. Literature shows that multiparous women often have higher confidence levels than primiparous women due to their previous experience(s), which enhances their participation in decision making and, in turn, positively influences their birth experience [57]. As primiparous women do not have a previous experience to build upon, it can be helpful to pay extra attention to their personal wishes and expectations during pregnancy. Furthermore, it is important to prepare primiparous women for the unpredictability of labour in order to strengthen their role during labour and birth.

Lower quality of experienced respect were found among women with a high education level compared to women with a low education level, which is in contrast with results of previous studies [16, 17]. However, Baranowska et al. (2019) found that women with a high education level reported more concerns regarding information provision and informed consent during labour and birth compared to women with a low education level [58]. It is possible that women with a high education level have higher expectations of care provision and are more aware of their rights, making them more critical towards interaction with their care provider. Furthermore, the general role of being a patient is changing. While previously caregivers played a dominant role in decision-making in patients' health care, today patients want to be more engaged [59]. Higher educated patients may be forerunners when it comes to this transition. It is important for care providers to stimulate active patient engagement so their care provision can be more in line with their client's preferences and needs.

## Strengths and limitations

To our knowledge, this is the first study specifically focusing on experienced client-care provider interaction during labour and birth in the Netherlands. A large number of women from four different regions in the Netherlands participated in the study. The measurement instrument used was a validated questionnaire, developed to evaluate the maternity health care system in the Netherlands. The questionnaire was based on the WHO responsiveness concept, which aims to measure the quality of care from a client's perspective, protect human rights in

pregnancy and childbirth and to optimise birth outcomes. By presenting the results of the four domains in this study, more insight is given in specific types of interaction and related factors.

The questions and domains were dichotomised in optimal and non-optimal interaction, with most of the questions only the answer option 'always' considered as optimal interaction, leaving the answer options 'never', 'sometimes' and 'often' to represent non-optimal interaction. Although this is a strict categorisation, striving for optimal interaction should be the standard in maternity care, especially in a high quality health care setting such as the Netherlands. For three questions, the answer option classification was not predefined and needed to be decided upon by the researchers. This could have led to some form of subjective interpretation. However, the research team reached consensus about the classification of the answer options as optimal or non-optimal with input from two clients.

Previous literature shows a broad time range in which studies on childbirth experience take place, however there is insufficient evidence on the optimal moment of measurement [3, 5]. Waldenström (2004) showed that studies performed soon after childbirth can lead to more positive response as there is a sense of relief that childbirth is over. Negative aspects often take longer to integrate [60]. Another study by Waldenström (2004) showed that in women who completed a questionnaire after two months compared to after one year, the majority described the same experience. However, there was also a group of women who looked back more negatively one year postpartum compared to two months postpartum [61]. Hodnett (2002) argued that the optimal time for questionnaires depends on the purpose of the study and that other aspects, such as guaranteeing the privacy of the respondent in order to prevent socially desirable answers or gratitude bias is more important for the validity of the study than timeframe [5]. In the current study, respondents were invited to fill in a questionnaire by email two months after birth, giving respondents the opportunity to share their experience after the postpartum period in their own preferred setting within a time range of four weeks.

The study population consisted of more ethnic Dutch and more highly educated women compared to the general Dutch population. It is possible that the group of women who were invited to participate as a whole was a better reflection of the Dutch population. As all data on characteristics were collected through the questionnaire we did not have access to the characteristics of the non-responders. Furthermore, the variable of self-identified ethnicity did not provide information on the respondents race. As race plays a role in health care provision [62], this needs to be taken into account when interpreting this variable. The questionnaire was only available in Dutch and had to be completed online. Therefore, women who could not read Dutch or who did not have access to the internet were not able to participate in the study. Even though in our study lower educated groups reported higher quality of respect, previous research shows that a different ethnic background and a lower education level are associated with a lower quality of experienced respectful care [15]. Therefore, it is important to take the underrepresentation of these groups into account when interpreting the results of the current study, specifically the underrepresentation of migrant women and women with a refugee background. More primiparous women and women who gave birth spontaneously and at home were included. The recruitment procedure through midwifery practices in the Netherlands could explain the higher number of spontaneous home births, as women with a high-risk pregnancy receive care from obstetricians in the hospital and were therefore less likely to be recruited for the study during pregnancy. Some of the variable categories consisted of small numbers of respondents (e.g. for age and mode of birth), which should be taken into account when interpreting the results.

Lastly, the current study only includes the client's perspective on client-care provider interaction. As the interaction between clients and care providers includes two parties, it is important to also study the care providers point of view.

## Conclusion

This study shows that on average women scored high on experienced client-care provider interaction in the domains respect, communication, confidentiality and autonomy. At the same time, client-care provider interaction in the Netherlands still fell short of being optimal for a large number of women. This indicates that there is room for improvement, in particular regarding women's autonomy during labour and birth. Women who gave birth at home experienced more optimal client-care provider interaction compared to other birth settings. These results show the need for attention on client-care provider interaction during labour and birth, hereby securing Respectful Maternity Care provision for all women, regardless of birth setting.

## Supporting information

**S1 Table. Univariate (OR) and multivariate logistic regression (AOR) models to assess variables associated with optimal interaction in the domains respect and communication.** (DOCX)

**S2 Table. Univariate (OR) and multivariate logistic regression (AOR) models to assess variables associated with optimal interaction in the domains confidentiality and autonomy.** (DOCX)

## Author Contributions

**Conceptualization:** Marit S. G. van der Pijl, Marlies Kasperink, Martine H. Hollander, Corine Verhoeven, Elselijn Kingma, Ank de Jonge.

**Data curation:** Marit S. G. van der Pijl, Corine Verhoeven, Ank de Jonge.

**Formal analysis:** Marit S. G. van der Pijl, Marlies Kasperink.

**Investigation:** Marit S. G. van der Pijl, Marlies Kasperink, Martine H. Hollander, Corine Verhoeven, Elselijn Kingma, Ank de Jonge.

**Methodology:** Marit S. G. van der Pijl, Marlies Kasperink, Martine H. Hollander, Corine Verhoeven, Elselijn Kingma, Ank de Jonge.

**Project administration:** Marit S. G. van der Pijl, Corine Verhoeven, Ank de Jonge.

**Supervision:** Martine H. Hollander, Corine Verhoeven, Elselijn Kingma, Ank de Jonge.

**Writing – original draft:** Marit S. G. van der Pijl, Marlies Kasperink, Martine H. Hollander, Corine Verhoeven, Elselijn Kingma, Ank de Jonge.

**Writing – review & editing:** Marit S. G. van der Pijl, Marlies Kasperink, Martine H. Hollander, Corine Verhoeven, Elselijn Kingma, Ank de Jonge.

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
