## [Decision Letter · Decision Letter 0]

17 Dec 2020

PONE-D-20-27732

Client-care provider interaction during labour and birth as experienced by women: respect, communication, confidentiality and autonomy.

PLOS ONE

Dear Dr. van der Pijl,

Thank you for submitting your manuscript to PLOS ONE. After careful consideration, we feel that it has merit but does not fully meet PLOS ONE’s publication criteria as it currently stands. Therefore, we invite you to submit a revised version of the manuscript that addresses the points raised during the review process.

We look forward to receiving your revised manuscript.

Kind regards,

Antonio Simone Laganà, M.D., Ph.D.

Academic Editor

PLOS ONE

Additional Editor Comments:

The reviewers have expressed positive comments regarding your article, raising only few concerns. Considering this point, I invite authors to perform the required minor revisions.

Journal Requirements:

3. Please include a copy of Table 5 which you refer to in your text on page 18.

Reviewers' comments:

Reviewer's Responses to Questions

**Comments to the Author**

1. Is the manuscript technically sound, and do the data support the conclusions?

Reviewer #1: Partly

Reviewer #2: Partly

Reviewer #3: Yes

2. Has the statistical analysis been performed appropriately and rigorously? 

Reviewer #1: Yes

Reviewer #2: Yes

Reviewer #3: I Don't Know

3. Have the authors made all data underlying the findings in their manuscript fully available?

Reviewer #1: Yes

Reviewer #2: Yes

Reviewer #3: Yes

4. Is the manuscript presented in an intelligible fashion and written in standard English?

Reviewer #1: Yes

Reviewer #2: Yes

Reviewer #3: Yes

5. Review Comments to the Author

Reviewer #1: Thank you for the opportunity to review this paper. The topic is important and I commend the authors for addressing this area.

I have a number of issues to raise.

The Abstract needs to give the country of where the study was undertaken to give the reader immediate context. This also needs to occur in the Introduction section.

The Abstract states that the domains were dichotomized to undertake the logistic regression. Is this a validated and accepted approach? What cut offs were used to do this? The results suggest it was 3.5 – why was this selected?

Were all women who filled in the survey included. The Abstract states that 767 were included – was that everyone or was a sample selected?

The abstract conclusion states that the majority of women experienced high quality care which is correct but it was actually less than two thirds – more than one third of women reported non-optimal interactions which is concerning.

Women who could not read Dutch were not included. What do the authors feel would have been the responses if women from migrant and refugee backgrounds who did not speak Dutch would have been? It is likely that the results would have been lower and this might warrant a comment in the Discussion.

The respondents were different on almost every characteristic (Table 2). Why was that and what are the implications for the generalizability?

I found Table 3 hard to follow. I understand the overall n and % for each domain and the n and % for the likert scale results (col 3). I don’t follow Col 4 for the likert scale responses. This is one n and % from 5 responses? Equally, for example, the overall % of optimal care for the domain (eg Respect) was 53% but each of the domain elements below this are 75% and above. How did the 53% get worked out? Maybe I am not reading this correctly but the table needs for clarity. This si the same for all of table 3.

Some of the writing is hard to follow and needs some editing. For example – women who gave birth with a community midwifery had optimal scores more often than women …… could be rephrased to be: women who gave birth with a community midwifery a high proportion of optimal scores than women …… There are a number of typographical and spelling errors (eg. line 61 and 72). PSTD needs to be written out the first time it is used (line 62).

Reviewer #2: I was pleased to revise the manuscript entitled “Client-care provider interaction during labour and birth as experienced by women: respect, communication, confidentiality, and autonomy.” (Manuscript Number: PONE-D-20-27732).

In my honest opinion, the topic is interesting enough to attract the readers’ attention. The methodology is accurate, and the data analysis supports conclusions. Nevertheless, the authors should clarify some points, as suggested below:

- I would suggest checking the use of British and American English in the text and tables (i.e., the use of both cesarean and caesarean).

- Regarding study limitations, the main concern regards the high risk of recall bias. Questionnaires were sent two months after delivery.

- I would suggest clarifying the source of delivery details, such as the induction of labor and delivery mode. Were details asked the patients or retrieved by medical records?

- The response rate should be discussed. Are the characteristics of women who did not respond available?

- Episiotomy appears as an important factor. In this regard, I would suggest discussing more, in general, the importance of avoiding routine use of episiotomy favoring a selective use. Refer to: PMID: 31823037

- Unplanned cesarean section is reported as a factor associated with communication score. Were cesarean section complications considered as a factor influencing such association? Cesarean section is major surgery, and associated complications may have influenced this observed association. Refer to: PMID: 30877907; PMID: 28681107

Reviewer #3: Manuscript: Client-care provider interaction during labour and birth as experienced by women.

Summary: This manuscript share statistical analysis of a web based survey (ReproQ) to analyze people’s experiences of 4 domains known to impact quality of care: respect, autonomy, communication and confidentiality/privacy. This offers further evidence of the how the tenants of respectful maternity care are being experienced/or not experienced by women.

Intro:

Good narrative overview of current literature

Suggest that authors provide clear definitions of autonomy and disrespect. These terms are being universally applied but are culturally nuanced. Define it in the Dutch context. Having greater precision around these terms is critical to progress. What is mistreatment? What is respect? What is autonomy? I would guess that providers would also have a hard time defining this term – and the misalignment of values is where we do not direct enough of our analysis or critique of the issues related to RMC.

Pg3 (ln 65-70) Early on, authors mention the Vedam (2019) study and women of color , younger women and lower SES– however this analysis is not specify if women of color or income level. Suggest citing their broader statistic that is mentioned later in the manuscript (1 in 6 women experience mistreatment)

Pg4 (ln 80-82) authors state: “Although the literature does not provide one clear definition of interaction, it can be described as a 81 process of cognition and action, in which the actions can be physical acts, acts of interplay or contact 82 of verbal or nonverbal communication [19]” Agreed – however a true analysis of any interaction or relationship requires both parties are addressed. Please specify that this only analyzes the client perspective and perhaps in the limitation, it can include that analysis of provider perspectives is needed.

Consider including the history of RMC being rooted in a human rights framework – this grounds the work in a broader discourse.

Methods:

Recommend the word ‘tiers’ instead of echelons.

Define ‘secondary obstetrician-led units for international audience

Pg. 5(ln111-113) Provide rationale for this investment in collaborative practice? Why was this created?

Data analysis:

Most of my questions in the analysis is how the decision to dichotomize was made? What was the rationale for not including ‘often’ with ‘always’. This would seem a less skewed dichotomization, but authors did not choose this cutoff. Is there precedent for this cut point? Since your significance was found when using this dichotomization technique – it seems valuable for the readers to understand the rationale for this choice.

Pg9-10 (lns 203-228) is very particular and shows the thinking process around specific questions – can this be condensed?

Results

In the demographic table, you characterize clients as Dutch and non-Dutch. I would think Dutch characterizes your citizenship status? So, can’t someone be Turkish and Dutch? Are these mutually exclusive? This categorization, as an American, is odd. For example, you can be Dutch-American or Mexican-American. We categorize by race, which has its own problematics. However, I would suggest if we are talking about interaction in care – racism and xenophobia become critical factors. Every country suffers from racism, so my question is what are you trying to distinguish here? I think in this time of the global movement for Black Lives – it is really important that we, as researchers, apply our tools with precision.

For each domain, I would suggest adding a definition to each domain. It may seem tedious and unnecessary, but I think it would help readers and set standards for providers who will be the most likely folks reading this work.

Discussion

It seems overall, people in Holland, more often than not, report feeling respected. However, mistreatment is different than respect and that needs to be clarified here. When we start to use and conflate terms it makes the research less reliable. It would be particularly useful to provide a definition of autonomy and refer to bioethical literature on this concept as well.

It might also be helpful to cite some feminist principles and literature – while returning to the core human rights principles so readers can situation this work in the more critical conversation of why women are not being listened to, respected or are unable to assert their autonomy.

Strengths/Limitations

As stated above, clear rationale for how women were identified by their ethnic group is needed if we are to understand the deeper cultural roots of disrespect, mistreatment and limitation placed on one’s autonomy. The Dutch system is very unique in terms of your positive outcomes; however, the readership is global. Do not assume that there is a clear rationale for your demographic variables – explain this to us and try to apply an anti-racist frame to why women were categorized this way and why there was no analysis of non-Dutch speakers or refugee women – who may experience – high levels of disrespect and mistreatment.

6. PLOS authors have the option to publish the peer review history of their article (what does this mean?). If published, this will include your full peer review and any attached files.

Reviewer #1: No

Reviewer #2: No

Reviewer #3: **Yes: **P. Mimi Niles

---

## [Author Response · Author response to Decision Letter 0]

20 Jan 2021

Please find a detailed response to the comments in the document ‘Response to Reviewers’.

---

## [Decision Letter · Decision Letter 1]

25 Jan 2021

Client-care provider interaction during labour and birth as experienced by women: respect, communication, confidentiality and autonomy.

PONE-D-20-27732R1

Dear Dr. van der Pijl,

We’re pleased to inform you that your manuscript has been judged scientifically suitable for publication and will be formally accepted for publication once it meets all outstanding technical requirements.

Kind regards,

Antonio Simone Laganà, M.D., Ph.D.

Academic Editor

PLOS ONE

Additional Editor Comments (optional):

Authors performed the required corrections, which were positively evaluated by the reviewers. I am pleased to accept this paper for publication.

Reviewers' comments:

Reviewer's Responses to Questions

**Comments to the Author**

1. If the authors have adequately addressed your comments raised in a previous round of review and you feel that this manuscript is now acceptable for publication, you may indicate that here to bypass the “Comments to the Author” section, enter your conflict of interest statement in the “Confidential to Editor” section, and submit your "Accept" recommendation.

Reviewer #1: All comments have been addressed

Reviewer #2: All comments have been addressed

2. Is the manuscript technically sound, and do the data support the conclusions?

Reviewer #1: Yes

Reviewer #2: Yes

3. Has the statistical analysis been performed appropriately and rigorously? 

Reviewer #1: Yes

Reviewer #2: Yes

4. Have the authors made all data underlying the findings in their manuscript fully available?

Reviewer #1: Yes

Reviewer #2: Yes

5. Is the manuscript presented in an intelligible fashion and written in standard English?

Reviewer #1: Yes

Reviewer #2: Yes

6. Review Comments to the Author

Reviewer #1: (No Response)

Reviewer #2: I was pleased to revise the manuscript entitled “Client-care provider interaction during labour and birth as experienced by women: respect, communication, confidentiality, and autonomy.” (Manuscript Number: PONE-D-20-27732).

In my honest opinion, the topic is interesting enough to attract the readers’ attention. The methodology is accurate, and the data analysis supports conclusions. Moreover, the authors addressed all suggested revisions, and I appreciated the manuscript improvement.

7. PLOS authors have the option to publish the peer review history of their article (what does this mean?). If published, this will include your full peer review and any attached files.

Reviewer #1: **Yes: **Caroline Homer

Reviewer #2: No

---

## [Editor Report · Acceptance letter]

3 Feb 2021

PONE-D-20-27732R1 

Client-care provider interaction during labour and birth as experienced by women: respect, communication, confidentiality and autonomy. 

Dear Dr. van der Pijl:

I'm pleased to inform you that your manuscript has been deemed suitable for publication in PLOS ONE. Congratulations! Your manuscript is now with our production department. 

Kind regards, 

on behalf of

Dr. Antonio Simone Laganà 

Academic Editor

PLOS ONE